# Epitope–Paratope Interaction of a Neutralizing Human Anti-Hepatitis B Virus PreS1 Antibody That Recognizes the Receptor-Binding Motif

**DOI:** 10.3390/vaccines9070754

**Published:** 2021-07-07

**Authors:** Jisu Hong, Youngjin Choi, Yoonjoo Choi, Jiwoo Lee, Hyo Jeong Hong

**Affiliations:** 1Department of Systems Immunology, College of Biomedical Science, Kangwon National University, Chuncheon 24341, Korea; ghdwltn55@kangwon.ac.kr (J.H.); skdmltmxk1@kangwon.ac.kr (Y.C.); snm04062@kangwon.ac.kr (J.L.); 2Medical Research Center, Chonnam National University Medical School, Hwasun 58128, Korea; yoonjoo.choi@jnu.ac.kr

**Keywords:** hepatitis B virus, human monoclonal antibody, virus entry inhibitor, PreS1, epitope, paratope, receptor-binding motif

## Abstract

Hepatitis B virus (HBV) is a global health burden that causes acute and chronic hepatitis. To develop an HBV-neutralizing antibody that effectively prevents HBV infection, we previously generated a human anti-preS1 monoclonal antibody (1A8) that binds to genotypes A–D and validated its HBV-neutralizing activity in vitro. In the present study, we aimed to determine the fine epitope and paratope of 1A8 to understand the mechanism of HBV neutralization. We performed alanine-scanning mutagenesis on the preS1 (aa 19–34, genotype C) and the heavy (HCDR) and light (LCDR) chain complementarity-determining regions. The 1A8 recognized the three residues (Leu22, Gly23, and Phe25) within the highly conserved receptor-binding motif (NPLGFFP) of the preS1, while four CDR residues of 1A8 were critical in antigen binding. Structural analysis of the epitope–paratope interaction by molecular modeling revealed that Leu100 in the HCDR3, Ala50 in the HCDR2, and Tyr96 in the LCDR3 closely interacted with Leu22, Gly23, and Phe25 of the preS1. Additionally, we found that 1A8 also binds to the receptor-binding motif (NPLGFLP) of infrequently occurring HBV. The results suggest that 1A8 may broadly and effectively block HBV entry and thus have potential as a promising candidate for the prevention and treatment of HBV infection.

## 1. Introduction

Hepatitis B virus (HBV), a small-enveloped DNA virus, infects human hepatocytes and causes acute and chronic hepatitis B, which results in a high risk of developing cirrhosis and hepatocellular carcinoma [1,2]. The number of worldwide chronic carriers in 2016 was estimated to be more than 290 million people [3]. To date, at least eight HBV genotypes (A–H) with a distinct geographic distribution have been identified [4,5,6,7]. Genotypes A and D are ubiquitous but most prevalent in Africa and Europe, while genotypes B and C are prevalent in Asia and Oceania; genotypes E–H are confined to Asia. Genotypes A–D constitute approximately 90% of total hepatitis B patients [8], whereas genotypes E–H represent less than 5% [8,9].

The HBV envelope contains three membrane proteins—large (L), middle (M), and small (S) proteins [10,11]. The S protein is the common C-terminal domain of the three envelope proteins, the M protein contains an N-terminal preS2 domain and the S domain, and the L protein contains an N-terminal preS1 domain and the preS2 and S domains [11]. The length of the preS1 domain is 119 (genotypes A–C, F, H, and I), 118 (genotypes E and G), or 108 amino acids (aa) (genotypes D and J) [4,11]. When HBV infects hepatocytes, amino acids (aa) 2–47 (genotype D) in the preS1 are critical for the interaction of a viral particle with an HBV receptor, sodium taurocholate cotransporting polypeptide (NTCP), while a short and highly conserved motif (NPLGFFP at aa positions 9–15 in genotype D) is essential for attachment and subsequent infection [12,13]. In addition to viral particles, HBV-infected hepatocytes also produce a large number of non-infectious subviral particles that consist mainly of the S protein, which have been suspected to reduce the virus-specific immune response by mimicking virions [14,15].

Hepatitis B immune globulins (HBIGs), prepared from donors’ plasma containing anti-S antibodies, have proven useful for passive prophylaxis against HBV infection. HBIGs neutralize circulating viral particles and thus are administrated to prevent mother–child vertical transmission and the reinfection of liver transplants in patients with chronic HBV-related liver disease [16,17]. However, HBIGs are not an ideal source of antibodies due to their limited availability and low specific activity [18]. Therefore, the development of HBV-neutralizing monoclonal antibodies (mAbs) with high efficacy is inevitable. Moreover, given that the preS1 domain plays an essential role in HBV entry, the generation of neutralizing anti-preS1 mAbs that can block viral entry into hepatocytes would be an effective strategy for the prevention and treatment of HBV infection.

We previously generated a human mAb (1A8) to the preS1 of HBV from a human synthetic Fab library and confirmed that it binds to the virus particles (genotype D) and neutralizes HBV infection in an in vitro neutralization assay using HepG2 cells overexpressing NTCP [19]. In the present study, we aimed to understand the mechanism of HBV neutralization. We determined the fine epitope and paratope of 1A8 through alanine-scanning mutagenesis on the preS1 (aa 19–34, genotype C) residues and the HCDR and LCDR residues of 1A8, respectively. Regarding the results, 1A8 recognized the three residues (Leu22, Gly23, and Phe25) within the receptor-binding motif (NPLGFFP), while four CDR residues of 1A8 were critical in antigen binding. Next, we performed structural analysis of the epitope–paratope interaction by molecular modeling. The results revealed that Leu100 in the HCDR3 and Tyr96 in the LCDR3 closely interacted with Leu22 and Gly23 of the preS1, while Ala50 in the HCDR2 interacted with Phe25 of the preS1. In addition, we demonstrated that 1A8 can bind to the preS1 antigen of genotypes A–D with the conserved receptor-binding motif and also to the preS1 with the receptor-binding motif (NPLGFLP), carrying a leucine substitution of Phe25 [20]. The results suggest that 1A8 may broadly and effectively block HBV entry and thus may have potential as a promising candidate for the prevention and treatment of HBV infection.

## 2. Materials and Methods

### 2.1. Cells and Cell Culture

HEK293F cells (Invitrogen, Thermo Fisher Scientific, Waltham, MA, USA) were cultured in FreeStyle 293 Expression medium (Gibco, Thermo Fisher Scientific, Waltham, MA, USA) at 37 °C and 125 rpm in a humidified incubator with 8% CO_2_.

### 2.2. Alanine Scanning Mutagenesis of PreS1

The gene encoding the preS1 (aa 1–56) of genotype C [21] fused to Strep-tag II (WSHPQFEK) was subcloned into the *Bam*HI–*Eco*RI sites of pGEX-2T (Pharmacia) to construct pGST–preS1 (aa 1–56)–strep expression plasmid. Alanine-scanning mutagenesis on the preS1 (aa 19–34) residues was performed by recombinant PCR using pGST–preS1 (aa 1–56)–strep DNA as a template and mutagenic primers. The final PCR products were digested with *Bam*HI and *Eco*RI (NEB) and individually subcloned into pGEX-2T.

### 2.3. Preparation of Recombinant PreS1 Antigen

The pGST–preS1 (aa 1–56)–strep plasmid DNA carrying the wild-type or alanine substitution was introduced into *Escherichia coli* DH5α and expression was induced by 0.2 mM IPTG at 18 °C for 19 h [22,23]. For purification of the GST–preS1 (aa 1–56)–strep protein, the induced cells were harvested, and the cell lysates were prepared by sonication. The supernatant was recovered and applied to a glutathione-agarose column (GenScript, Piscataway, NJ, USA). The bound antigen was eluted according to the manufacturer’s protocol. The protein concentration was determined using a Nano-Drop 2000 (Thermo Fisher Scientific, USA). The purity of the recombinant preS1 antigen was assessed by 12% sodium dodecyl sulfate–polyacrylamide gel electrophoresis (SDS–PAGE).

### 2.4. Mutagenesis, Expression, and Purification of Antibodies

Alanine-scanning mutagenesis or site-directed mutagenesis was performed on the HCDR (HCDR1, HCDR2, or HCDR3) or LCDR3 residues of 1A8 by recombinant PCR using the heavy or light chain expression plasmid for 1A8 as a template and mutagenic primers. The mutated VH or VL gene was digested with restriction enzymes and subcloned into the *Eco*RI*-Apa*I or *Hin*dIII*-Bsi*WI sites of pCMV-dhfr containing human Cγ1 or Cκ, respectively.

The resulting heavy and light chain expression plasmids were introduced into 30 mL of HEK293F cells (1 × 10^6^ cells/mL) using polyethyleneimine (Polysciences, Warrington, PA, USA) at a ratio of 1:4 (30:120 µg) and cultured for seven days. For antibody purification, the culture supernatants were applied to affinity chromatography using Protein A-agarose beads (Amicogen, Inc., Gyeongsangnam-do, Korea) as described previously [19,24]. The purified antibody was stored at 4 °C in the Dulbecco’s phosphate-buffered saline (DPBS, Welegene). The antibody concentration was determined by a Nano-Drop 2000 (Thermo, USA), and the integrity and purity of the purified antibody were assessed by SDS–PAGE. 

### 2.5. ELISAs

A quantitative ELISA was performed to determine the concentration of antibodies. Anti-human IgG kappa antibodies (Thermo Fisher Scientific) were coated on a 96-well Maxisorp plate (100 ng/well) at 4 °C overnight and incubated with a serially diluted antibody in 0.1% PBA at 37 °C for 1 h. The bound antibody was detected with anti-human IgG (Fc-specific)-HRP antibodies (1:10,000, Thermo) as described previously [24]. 

Another quantitative ELISA was performed to determine the concentration of purified recombinant preS1 proteins. Serially diluted GST–preS1 (aa 1–56)–strep was coated on the 96-well plate at 4 °C overnight and incubated with mouse anti-strep tag II antibodies (200 ng/well, IBA). The bound antibodies were detected with anti-mouse IgG (Fc-specific)-HRP antibody (1:10,000, Thermo) as described above. 

An indirect ELISA was performed to determine the antigen-binding activity of the antibodies. The preS1 antigen coated on the 96-well plate (30 nM/well) was incubated with a serially diluted antibody. The bound antibodies were detected with anti-human IgG (Fc-specific)-HRP antibodies (1:10,000, Thermo), as described above.

### 2.6. Affinity Determination of Antibodies by Bio-Layer Interferometry (BLI)

Antibody affinity was determined by BLI using Octet Red384, as described previously [19], except that the association and dissociation rates were measured for 5 and 10 min, respectively. 

### 2.7. Computational Structural Modeling

The antibody structure was constructed using ABodyBuilder with MODELLER [25]. The final structure was further energy-minimized using the Tinker molecular dynamics package [26] with AMBER99sb [27] and the GB/SA solvent model [28]. The linear epitope region for the structural analysis was 18-VPNPLGFFPDH-28. BLAST against the PDBAA database using BLOSUM45 was employed to identify template structures. Any identified structures with no atom coordinate information were excluded. The longest sequence match found was from 1T90 chain A 439–447 (VPAPMAFFP). The last two residues (DH) were manually added using Tinker and energy-minimized as described above. The two molecules were docked using the ClusPro web server [29]. Short molecular dynamics simulations were performed using Tinker with the same force field parameters as in the minimization at 310 K for 10 ns.

## 3. Results

### 3.1. Epitope Mapping of 1A8 Antibody

We previously generated a human mAb (1A8) by panning of a phage-displayed human synthetic Fab library against recombinant preS1 antigens, including the preS1 peptide (aa 20–32, genotype C) shown in Figure 1A. In the present study, to determine the fine epitope of 1A8, we substituted each of the residues in the preS1 (aa 19–34) of genotype C to alanine, constructed a series of expression plasmids of pGST–preS1 (aa 1–56)–strep carrying an alanine substitution at each position, and expressed the GST–preS1 fusion proteins in *E. coli*. The GST–preS1 (aa 1–56)–strep proteins of the 15 mutants and the wild-type were purified and subjected to 12% SDS–PAGE to confirm the quality of purified proteins (Figure 1B). Next, the antibody-binding activities of the 16 GST–preS1 proteins were analyzed by indirect ELISA and quantitative ELISA (Appendix A). The results revealed that the GST–preS1 (aa 1–56)–strep carrying L22A, G23A, or F25A substitution completely lost its binding activity to 1A8 (Figure 2), indicating that the fine epitope of 1A8 is Leu22, Gly23, and Phe25 of the preS1. 

### 3.2. Alanine-Scanning Mutagenesis of HCDR3 and LCDR3 of 1A8

HCDR3 and LCDR3 are formed by the junction of V–(D)–J and V–J joining, respectively, and therefore they are the most diverse among the six CDRs that typically form the antigen-binding surface [30,31]. To determine the critical residues in the HCDR3 and LCDR3 of 1A8 in antigen binding, we performed alanine-scanning mutagenesis on the 10 residues in the HCDR3 (aa 95–100) and LCDR3 (aa 91–96). The resulting alanine-replacement mutants of 1A8 were transiently expressed in HEK293F cells for seven days. Then, the culture supernatants were subjected to indirect ELISA (Figure 3A,B) using the GST–preS1 (aa 1–56)–strep proteins and quantitative ELISA (Figure 3C,D). The results revealed that two mutants (L100A in the HCDR3 and Y96A in the LCDR3) exhibited no or very low antigen-binding activity, whereas the other six mutants showed the same antigen-binding activities as 1A8. Two HCDR3 mutants (I96A and Y97A) were not expressed and therefore were exempted from the analysis. The results indicate that Leu100 in the HCDR3 and Tyr96 in the LCDR3 are in the paratope of 1A8, which directly contact the epitope.

### 3.3. Site-Directed Mutagenesis of HCDR1 and HCDR2 of 1A8

Position 33 in the HCDR1 and position 50 in the HCDR2 are the most diverse and mostly involved in direct contact with antigens [32,33]. To investigate whether these two residues contribute to antigen binding, we performed site-directed mutagenesis on Ala33 and Ser50 of 1A8. We generated eight mutants with serine, proline, valine, threonine, glycine, aspartic acid, leucine, or asparagine, instead of Ala33, as well as six mutants with alanine, tyrosine, glutamic acid, phenylalanine, valine, or proline, instead of Ser50. The mutants were transiently expressed in HEK293F cells, and the culture supernatants were analyzed by indirect and quantitative ELISAs.

In the case of the mutation at Ala33, three mutants (A33S, A33V, and A33P) exhibited the same antigen-binding activities as 1A8, whereas the other mutants (A33T, A33G, A33D, A33L, or A33N) exhibited no or decreased activities compared to 1A8 (Figure 4A–C). The results indicate that Ala33 may contribute to antigen binding, although A33S, A33V, or A33P mutation did not affect the antigen-binding activity of 1A8. In the case of the mutation at Ser50, the S50A mutant exhibited a slightly enhanced antigen-binding activity compared to 1A8, whereas three mutants (S50Y, S50F, and S50E) exhibited no or largely decreased activities (Figure 4D). Two mutants (S50V and S50P) were not expressed. The results indicate that Ser50 is critical for antigen binding. 

To examine whether Gln52a in the HCDR2 is involved in antigen binding, four mutants (Q52aA, Q52aF, Q52aM, and Q52aY) of Gln52a were constructed and transiently expressed in HEK293F cells. Analysis of the culture supernatants by ELISA revealed that the three mutants (Q52aA, Q52aF, and Q52aM) exhibited the same antigen-binding activities as 1A8, while the Q52aY mutant was not expressed (data not shown).

### 3.4. Affinity Determination

We previously determined the affinity (K_D_, 3.55 nM) of 1A8 for the preS1 by BLI using another recombinant preS1 antigen rather than the GST–preS1 fusion protein [19]. The recombinant preS1 antigen, L1 (Ig1–5)–preS1 (aa 1–56)–strep, was constructed by fusion of the preS1 (aa 1–60) of genotype A to the C-terminus of the Ig1–Ig5 domains of the human L1 cell adhesion molecule, while a strep-tag II was fused to the C-terminus of the preS1 (aa 1–60) [24]. Since we used GST–preS1 (aa 1–56)–strep in this study, we determined the affinity (K_D_) of 1A8 for freshly purified GST–preS1 (aa 1–56)–strep of genotype A by BLI with Octet Red384. The affinity of 1A8 for the GST–preS1 (aa 1–56)–strep was 0.50 nM (Table 1 and Appendix A), which was estimated to be 7-fold higher compared to that for L1 (Ig1–5)–preS1 (aa 1–56)–strep. This indicates that the affinity of 1A8 for the preS1 is influenced by the fusion partner.

### 3.5. Structural Analysis of the Antigen–Antibody Interaction by Molecular Modeling

Since we identified the fine epitope (Leu22, Gly23, and Phe25 of preS1) and paratope (Leu100 in the HCDR3, Tyr96 in the LCDR3, Ala33 in the HCDR1, and Ser50 in the HCDR2) of 1A8, we performed molecular modeling based on the fine epitope and paratope determined to predict the mechanism of antigen–antibody interaction. The structural model of 1A8 with Ala50 in the HCDR2 was built using ABodyBuilder. The model was constructed using a nearly sequence-identical structure (5VEB, sequence identity 99%), and all CDR structures were grafted using templates with high FREAD ESSS scores [34]. The structural model showed that the 1A8 CDRs form a narrow hole and Leu100 in HCDR3 is structurally buried (Figure 5A). 

Considering the alanine-scanning results of the epitope, we assumed that Phe24 may not directly interact with 1A8 (Figure 5B, left). To model the epitope peptide (18- VPNPLGFFPDH-28), BLAST against the PDBAA database was employed and the identified structure (1T90 chain A 439–447 VPAPMAFFP) was modified using Tinker based on the assumption (Figure 5B, right). 

The antibody and peptide were computationally docked using the ClusPro web server with the antibody mode, while attractions were assigned at the paratope and epitope residues. In total, 11 docking models were generated, but only one model matched the epitope and paratope profile. To further analyze detailed residue interactions, a short molecular dynamics simulation was performed at body temperature (310 K). The complex structure was quickly stabilized, and key residues were observed in close interactions. The docking model revealed that Leu100 in the HCDR3 and Tyr96 in the LCDR3 closely interacted with Leu22 and Gly23 of the preS1, while Ala50 in the HCDR2 interacted with Phe25 of the preS1 (Figure 5C). The major driving force between the two molecules seems to be hydrophobic interactions (Figure 5C).

Ala33 in the HCDR1 is solvent-exposed and may be in interaction with two residues (Phe24 and Asp27) in the preS1. Thus, mutations to other residues (A33T, A33G, A33D, A33L, or A33N) may lead to destabilization of binding, especially with the opposite charge (A33D). The structural analysis shows that substitutions with small hydrophobic amino acids, such as alanine, may be beneficial for binding, as verified in the mutagenesis study.

### 3.6. Analysis of the Antigen-Binding Activities of 1A8 to the Pres1 of Different HBV Genotypes

Since we identified that the fine epitope of 1A8 resides within the highly conserved receptor-binding motif, we investigated whether 1A8 can bind to the preS1 of all HBVs. We aligned the preS1 (aa 1–56) sequences of 9639 genes from eight genotypes (A–H) listed in the specialized HBV database [20] and compared their receptor-binding motif. As summarized in Table 2, most (9312 genes, 96.6%) HBVs had the conserved receptor-binding motif (NPLGFLP), but the genotype G and part (12.5%) of genotype F, as well as small percentages (0.5%–2.6%) of genotypes A–E, constituting 2.6% (247 genes) of the total genes, had a divergent motif (NPLGFLP) with Leu25 substitution for Phe25. The remaining 80 genes contained a heterogenous receptor-binding motif whose sequences were different from NPLGFFP or NPLGFLP.

Therefore, to analyze the antigen-binding activities of 1A8 to the genotypes A–D with the conserved receptor-binding motif, we expressed and purified the GST–preS1 (aa 1–56)–strep of genotypes A–D. Analysis of the purified antigens by indirect ELISA revealed that 1A8 bound to the preS1 antigens of genotypes A–D with the same affinity (Figure 6A).

Next, to analyze whether 1A8 can bind to the preS1 antigen with the divergent motif (NPLGFLP), we substituted Leu25 for Phe25 in the preS1 (aa1–56) of genotype A and expressed in *E. coli* as a GST fusion protein. The resulting GST–preS1 (aa 1–56)–strep (preS1-L25) protein was purified and analyzed by ELISA. The results showed that 1A8 bound to the preS1–L25, with a slightly decreased binding activity compared to the wild-type preS1 antigen with the conserved motif (Figure 6B). Similarly, the S50A mutant exhibited a slightly decreased binding activity to the preS1 antigen with the divergent motif compared to that with the conserved motif (Figure 6B). Given that 1A8 bound to the preS1 antigens of genotypes A–D with the same affinity, the results suggest that 1A8 and the S50A mutant may bind to all of the HBV genotypes (A–D) with the receptor-binding motif, NPLGFFP or NPLGFLP.

To investigate whether the docking model in Figure 5 can predict the difference in binding strength between the antibody (1A8 or the S50A) and antigen (preS1-F25 or preS1-L25), we built four computational models of 1A8 and preS1-F25 (S50-F25), S50-L25, S50A-F25, and S50A-L25 using FoldX (Ver. 5) [35], and calculated their interface energy values using the “AnalyzeComplex” command. The binding energy of S50A-F25 was strongest (-21.21). The binding energy differences against S50A-F25 show that the difference between S50A-L25 and S50-F25 is similar (ΔΔG: 0.96 and 0.9, respectively), while the binding affinity of S50-L25 was predicted the weakest (ΔΔG: 1.15). These data indicate that the computational results are consistent with the binding affinity data in Figure 6B, validating the structural model of the antigen–antibody interaction.

## 4. Discussion

HBIGs are polyclonal antibodies derived from pooled plasma containing anti-S antibodies and currently the only entry inhibitors to have obtained approval for clinical use [36]. They are only applied to prophylactic clinical contexts, such as preventing the reinfection of liver transplants in infected patients and preventing mother–child transmission [16,17,36]. HBIGs prevent HBV infection by binding to and neutralizing circulating virions. However, since NTCP was identified as a specific entry receptor for HBV, blocking viral entry into hepatocytes by targeting attachment steps has been explored as a possible strategy [36]. Several small molecular substances targeting NTCP, such as NTCP inhibitors and NTCP substrate inhibitors, have been tested for their ability to inhibit HBV uptake into hepatocytes, and some of them have been approved by the FDA [36]. To date, however, clinical development of the anti-preS1 mAb that blocks viral entry has not been reported.

In our previous attempt to develop a human mAb that blocks HBV entry into human hepatocytes, we generated a human mAb (1A8) that binds to the preS1 of genotypes A–D by panning a human synthetic Fab library against the recombinant preS1 antigens, including the peptide with the receptor-binding site, and confirmed its virus-neutralizing in an in vitro assay [19]. In the present study, to elucidate the mechanism of the HBV neutralization of 1A8, we determined the fine epitope and paratope of 1A8 through alanine-scanning mutagenesis and site-directed mutagenesis of both the antigen and antibody. The 1A8 recognized the three residues (Leu22, Gly23, and Phe25) within the highly conserved receptor-binding motif (NPLGFFP) of the preS1, and four CDR residues of 1A8 were critical in antigen binding. Subsequently, structural analysis of the epitope–paratope interaction by molecular modeling revealed that Leu100 in the HCDR3 and Tyr96 in the LCDR3 closely interacted with Leu22 and Gly23 of the preS1, while Ala50 in the HCDR2 interacted with Phe25 of the preS1. Additionally, we demonstrated that 1A8 exhibited binding activities to the two different receptor binding motifs (NPLGFFP and NPLGFLP) in the preS1 antigen, which were present in 99.2% of the total 9639 HBV genes listed in the Hepatitis B Virus Database, available online: https://hbvdb.lyon.inserm.fr/HBVdb/ (accessed on 28 May 2021). To the best of our knowledge, 1A8 is the first fully human anti-preS1 mAb that recognizes the receptor-binding motif of HBV. Taken together, the results suggest that 1A8 may broadly and effectively block HBV entry and thus have potential as a promising candidate for the prevention and treatment of HBV infection.

## 5. Conclusions

We successfully elucidated the mechanism of the epitope–paratope interaction by mutational analysis and structural analysis of the preS1 and 1A8 antibodies, which clearly showed that 1A8 binds to the fine epitope within the conserved receptor-binding motif in the preS1 of HBV. The 1A8 may have potential as an HBV entry inhibitor for the prevention and treatment of HBV infection.

## Figures and Tables

**Figure 1 vaccines-09-00754-f001:**
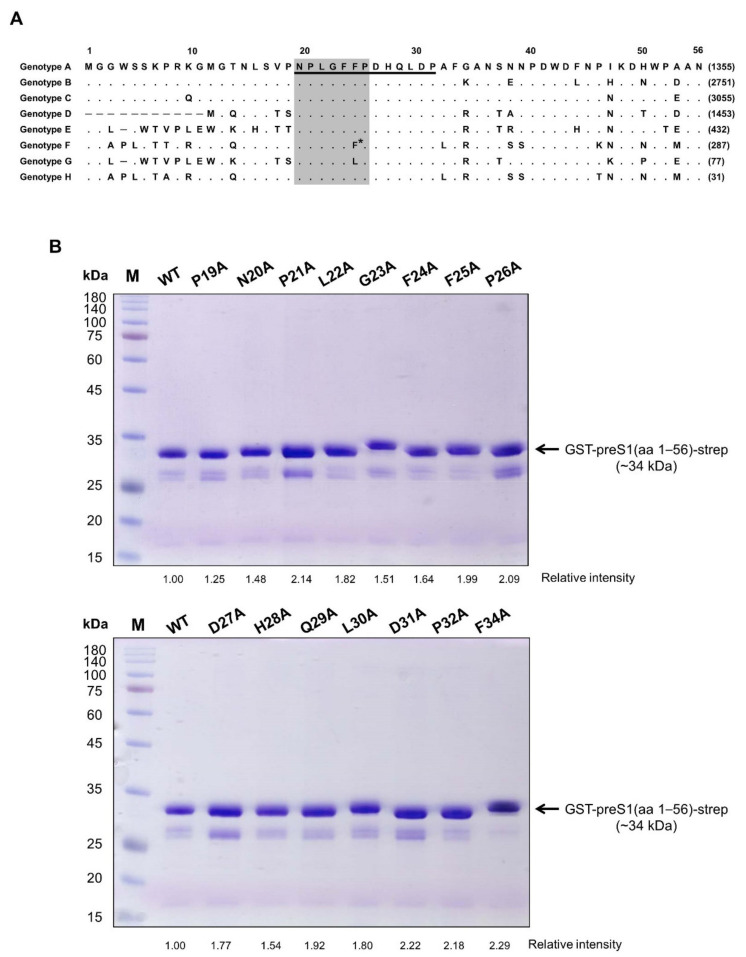
(**A**) Sequence alignment of preS1 (aa 1–56) of genotypes A–H. The number of identical sequences among the total 9639 preS1 genes in the Hepatitis B Virus Database, available online: https://hbvdb.lyon.inserm.fr/HBVdb/ (accessed on 28 May 2021) are written in parentheses after the amino acid sequences. A dot indicates an amino acid residue identical to that of genotype A, while a hyphen represents a deletion. In the case of genotype F, 81.53% and 12.47% of this genotype have Phe and Leu at position 25, respectively. The receptor-binding motif is shaded. (**B**) 12% SDS–PAGE of the purified GST–preS1 (aa 1–56)–strep with an alanine substitution at aa 19–34. The wild type (WT; genotype C) was included as a control. M, molecular weight standards.

**Figure 2 vaccines-09-00754-f002:**
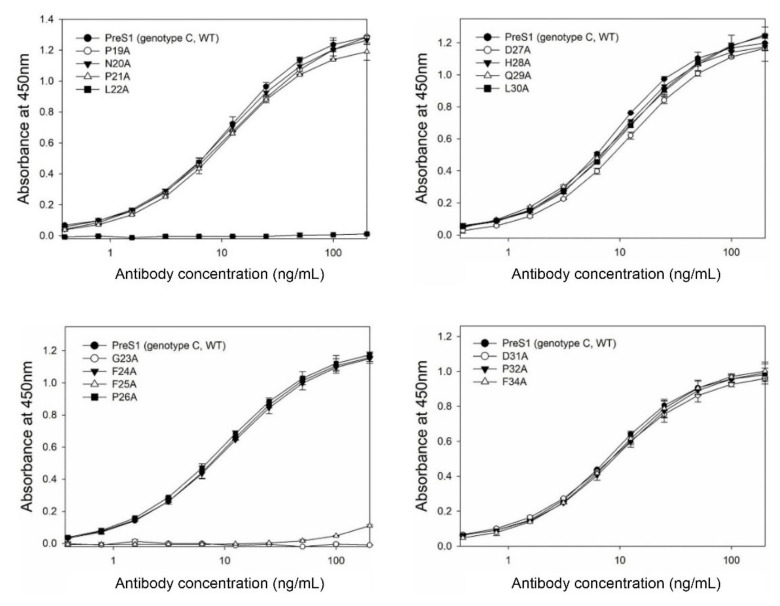
Analysis of the antibody-binding activities of purified GST–preS1 (aa 1–56)–strep carrying an alanine substitution by indirect and quantitative ELISAs (Appendix A). The wild-type (WT; genotype C) was included as a positive control. Values were obtained from duplicate wells and are expressed as the mean ± SD.

**Figure 3 vaccines-09-00754-f003:**
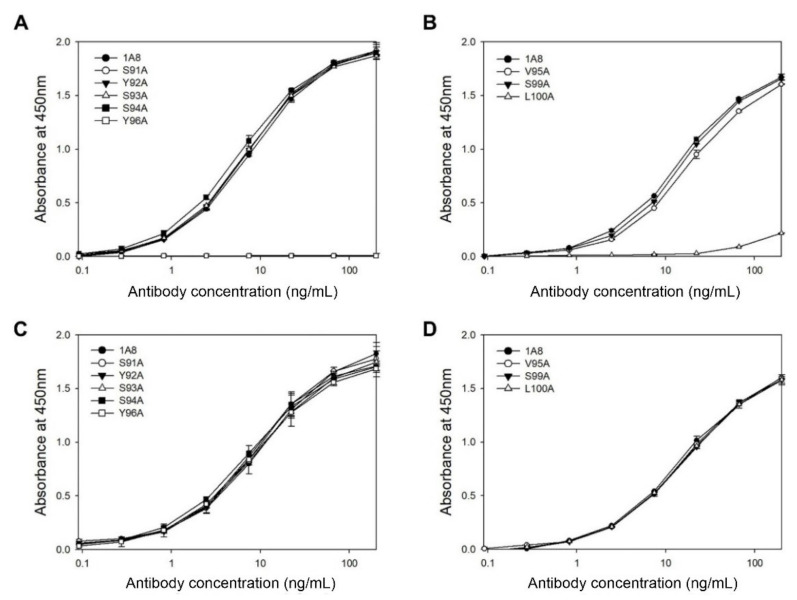
Analysis of the antigen-binding activities of 1A8 variants with an alanine substitution in the LCDR3 (**A**,**C**) or HCDR3 (**B**,**D**) by indirect ELISA (**A**,**B**) and quantitative ELISA (**C**,**D**). The culture supernatants containing each antibody (200 ng/mL) were serially diluted and subjected to the ELISAs. Values were obtained from duplicate wells and are expressed as the mean ± SD.

**Figure 4 vaccines-09-00754-f004:**
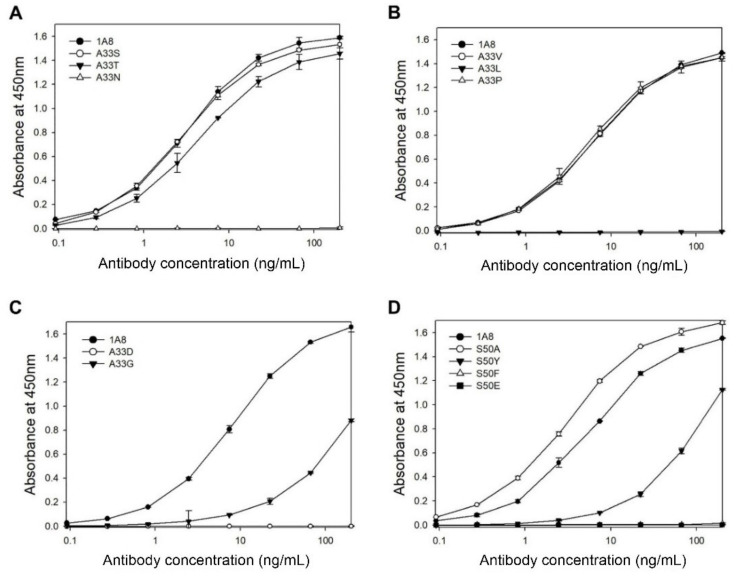
Analysis of the antigen-binding activities of 1A8 variants with a mutation at position 33 in the HCDR1 (**A**–**C**) or at position 50 in the HCDR2 (**D**) by indirect ELISA. The culture supernatants containing each antibody (200 ng/mL) was serially diluted and subjected to the ELISA. Values were obtained from duplicate wells and are expressed as the mean ± SD.

**Figure 5 vaccines-09-00754-f005:**
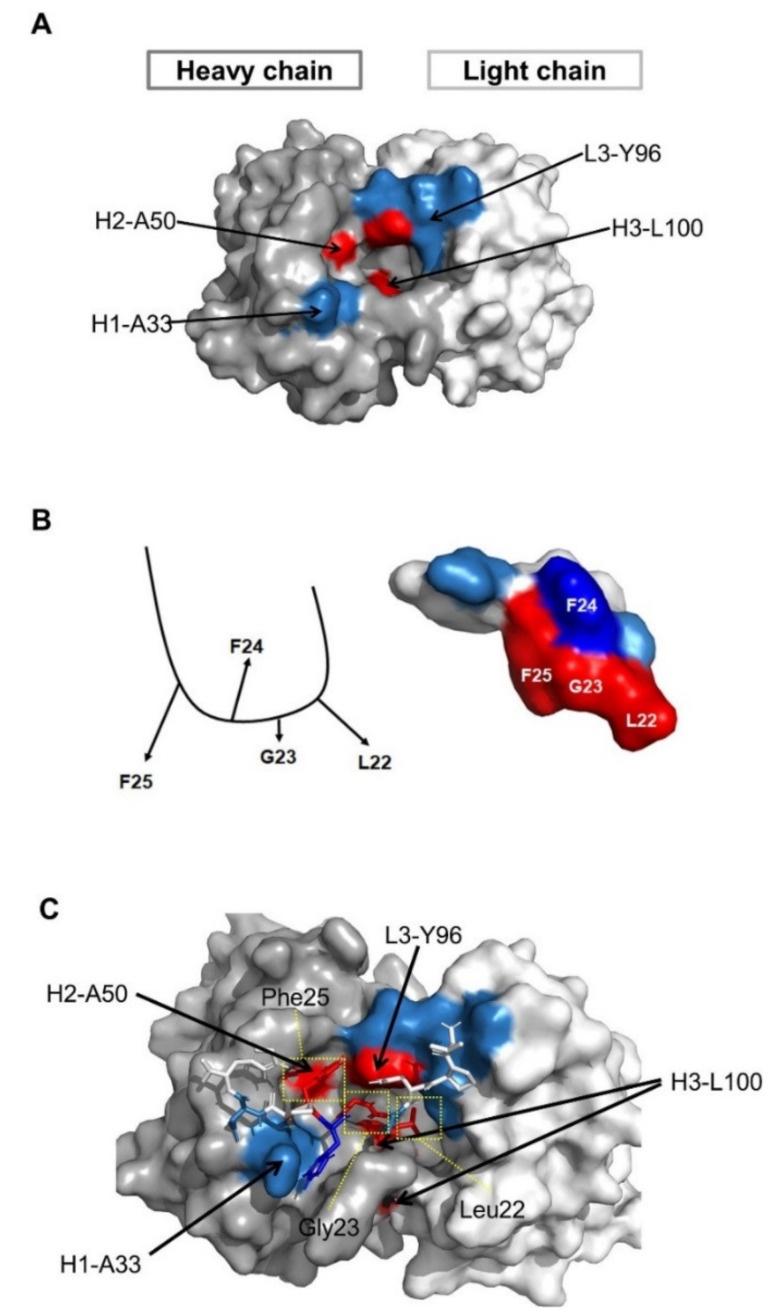
Structural analysis of the epitope–paratope interaction by molecular modeling. (**A**) Structural model of the VH (dark gray) and VL (light gray) of 1A8 with H2-A50. The residues that are essential for antigen binding are shown in red, and those that are less important or not involved in antigen binding are shown in blue. (**B**) Schematic diagram of the structure of the preS1 peptide. The three residues (Leu22, Gly23, and Phe25), shown in red, were identified as the fine epitope of 1A8. (**C**) Docking of the antigen and antibody models using the ClusPro method.

**Figure 6 vaccines-09-00754-f006:**
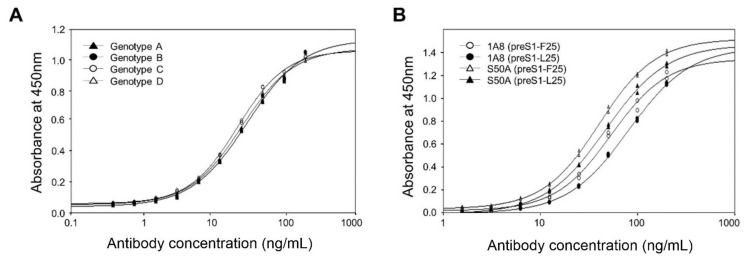
(**A**) Analysis of the antigen-binding activities of 1A8 to purified GST–preS1 (aa 1–56)–strep of genotypes A–D. The preS1 antigen (50 nM/well) was coated and incubated with a serially diluted antibody to perform indirect ELISA. (**B**) Analysis of the antigen-binding activities of 1A8 and the S50A mutant to purified GST–preS1 (aa 1–56)–strep of genotype A with Phe25 or Leu25. The preS1 antigen (100 ng/well) was coated and incubated with a serially diluted antibody to perform indirect ELISA. Values were obtained from duplicate wells and are expressed as the mean ± SD.

**Table 1 vaccines-09-00754-t001:** Affinity determination by Bio-Layer Interferometry with Octet Red384.

Antibody	K_D_	K_on_ (1/Ms)	K_dis_ (1/s)	Full R^2^
1A8	0.50 nM	3.92 × 10^5^	1.94 × 10^−4^	0.9969

Kon, rate of association; Kdis, rate of dissociation; Full R^2^, estimate of the goodness of the curve fit.

**Table 2 vaccines-09-00754-t002:** The numbers and percentages of the receptor-binding motif with different sequences in the genotypes A–H of HBV.

Receptor-Binding Motif (Number)	Genotype							
	A	B	C	D	E	F	G	H
NPLGFFP	1355	2751	3055	1453	432	234	1	31
(9312)	(96.50%)	(98.18%)	(97.73%)	(99.18%)	(96.86%)	(81.53%)	(1.28%)	(100%)
NPLGFLP	36	30	38	8	5	53	77	0
(247)	(2.56%)	(1.07%)	(1.22%)	(0.55%)	(1.12%)	(12.47%)	(98.72%)	(0.00%)
Others	13	21	33	4	9	0	0	0
(80)	(0.93%, 9 D.S. *)	(0.75%, 12 D.S.)	(1.06%, 18 D.S.)	(0.27%, 3 D.S.)	(2.02%, 8 D.S.)	(0.00%)	(0.00%)	(0.00%)
Total (9639)	1404	2802	3126	1465	446	287	78	31

* Different sequences.

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
