# Peer review of "Epitope–Paratope Interaction of a Neutralizing Human Anti-Hepatitis B Virus PreS1 Antibody That Recognizes the Receptor-Binding Motif"

_vaccines, 2021, doi:10.3390/vaccines9070754_

Round 1

Reviewer 1 Report

In this paper, Hong et al. performed a fine mapping of the interaction sites between anti-preS1 monoclonal antibody 1A8 and preS1 domain of HBV envelope protein. Using mutagenesis and ELISA, the authors clearly showed residues that are responsible for the binding. The results are important for the characterization of antibodies that may inhibit HBV infection. I have some specific comments to be addressed as below.

  1. Figure 1A. “F” at aa25 of genotype F is identical to genotype A. It should be a dot.
  2. Figure S1. For better understanding, please show which antibody was used in the figure legend.
  3. In each assay, please show how many samples were tested.
  4. Please show what error bars indicate in the figures (standard deviation or standard error?).
  5. Line 280. “A337” seems to be a mistake.

Author Response

  1. Figure 1A. “F” at aa25 of genotype F is identical to genotype A. It should be a dot. We included this sentence 'In the case of genotype F, 81.53% and 12.47% of this genotype have Phe and Leu at position 25, respectively(Table 2)' in the figure legend of Fig.1, because not all of genotype F have Phe at position 25.  
  2. Figure S1. For better understanding, please show which antibody was used in the figure legend. We added the following sentences 'Serially diluted GST-preS1(aa 1–56)-strep was coated on the 96-well plate at 4°C overnight and incubated with mouse anti-Strep tag II antibody (200 ng/well, IBA). Values were obtained from duplicate wells and are expressed as the mean ± SD' in the figure legend of Figure S1.
  3. In each assay, please show how many samples were tested. This sentence 'Values were obtained from duplicate wells and are expressed as the mean ± SD' was added to the figure legends of Figures 2-4, Figure 6, and Figure S1. 
  4. Please show what error bars indicate in the figures (standard deviation or standard error?). SD
  5. Line 280. “A337” seems to be a mistake. Changed to A33T

Reviewer 2 Report

Epitope-Paratope Interaction of A Neutralizing Human Anti-Hepatitis B Virus PreS1 Antibody That Recognizes the Receptor-Binding Motif

By Hong et al

Summary: The authors present a well organized and thorough characterization of the binding of a previously discovered antibody (1A8) known to bind to HBV preS1 and neutralizes viral infectivity. Counteracting HBV infection (prophylactically, treatment) is a very important goal and 1A8 offers promise as a therapeutic candidate for pursuing this goal. A detailed mechanism of how 1A8 works to neutralize HBV infection was not known prior to this study.

To further understand the mechanism of 1A8 neutralization the authors conduct positional scanning studies for both the antibody binding regions (complementarity determining regions, CDRs) and the epitope from preS1 HBV protein known to be bound by 1A8. The goal of these scans was to identify residues that are important for the antibody-preS1 interaction to generate a working model of how this interaction occurs. The authors use robust methodology, primarily relying on ELISA and molecular modeling to pinpoint important residues and generate a model of this interaction.

The experimental protocols and descriptions used in this manuscript appear robust and provide high quality data that leads to an informative model. Overall, this is a high-quality manuscript. It would benefit from making experimental predictions, and validating these predictions, from the model that is generated from the experimental data. This and other minor comments are provided below.

Major comments:

Does the model presented in Figure 5 provide any hypotheses that can be tested experimentally? It appears to suggest Ala50 from HCDR2 interacts with Phe25 of preS1. Several antibody variants in which Ala50 from HCDR2 is mutated are described. Viral variants in which Phe25 is mutated to Leu are also presented. This might suggest that Ala50 antibody variants might have an altered interaction with PreS1 F25L variant. If this were confirmed experimentally it would help to validate the value of the model presented in Figure 5.

Minor comments:

Page 1, line 14: there shouldn’t be two commas in “genotypes A-D,,”

Page 3 line 92: Protein expression was performed in DH5a bacteria. This is unusual as DH5a bacteria are typically poor hosts for protein expression. Can the authors comment on this?

Page 3, line 116: I am confused about the use of ELISA for quantitation of protein concentration (quantitative ELISA). When you plot quantitative ELISA data you have concentrations for the antibodies analyzed as ng/mL, but it seems that you are using the quantitative ELISA to determine concentration. Is there some kind of standard curve that is used to convert ELISA output to concentration? If so this should be described.

Figure 1 panel A. In the shaded receptor binding motif you have dots to indicate identical residues but for Genotype F you have “F” at position 25 even though this is identical to the Genotype A sequence. Shouldn’t this just be a dot?

Figure 2. Error bars are presented but not defined. Do error bars correspond to SD? Are these technical replicates? How many true replicates were performed for this experiment (and for other related experiments)?

Author Response

Major comments:

Does the model presented in Figure 5 provide any hypotheses that can be tested experimentally? It appears to suggest Ala50 from HCDR2 interacts with Phe25 of preS1. Several antibody variants in which Ala50 from HCDR2 is mutated are described. Viral variants in which Phe25 is mutated to Leu are also presented. This might suggest that Ala50 antibody variants might have an altered interaction with PreS1 F25L variant. If this were confirmed experimentally it would help to validate the value of the model presented in Figure 5.

   We built four computational models of 1A8 and preS1-F25 (S50-F25), S50-L25, S50A-F25, and S50A-L25 using FoldX (Ver. 5) and calculated their interface energy values using the "AnalyzeComplex" command. The results were consistent with expermental data, indicating that the structural model was validated (lines 309-318)

Minor comments:

Page 1, line 14: there shouldn’t be two commas in “genotypes A-D,,”

     A comma was deleted.

Page 3 line 92: Protein expression was performed in DH5a bacteria. This is unusual as DH5a bacteria are typically poor hosts for protein expression. Can the authors comment on this?

     Because we used E. coli DH5a in our previous studies, in the present study we used the same method for the expression of the GST-preS1 fusion protein.

Page 3, line 116: I am confused about the use of ELISA for quantitation of protein concentration (quantitative ELISA). When you plot quantitative ELISA data you have concentrations for the antibodies analyzed as ng/mL, but it seems that you are using the quantitative ELISA to determine concentration. Is there some kind of standard curve that is used to convert ELISA output to concentration? If so this should be described.

     We performed quantitative ELISA to confirm that the same amount of each different antibody sample was used to perform indirect ELISA. As seen in Figure 3C and 3D, the different antibody samples exhibited the same curves, indicating that their concentrations used in the indirect ELISA (Fig. 3A and 3B) were exactly same.

Figure 1 panel A. In the shaded receptor binding motif you have dots to indicate identical residues but for Genotype F you have “F” at position 25 even though this is identical to the Genotype A sequence. Shouldn’t this just be a dot?

    We changed from "F" to "F*" because in the case of genotype F, 81.53% and 12.47% of this genotype have Phe and Leu at position 25, respectively (as shown in Table 2). We added the sentence in the legend of Figure 1.

Figure 2. Error bars are presented but not defined. Do error bars correspond to SD? Are these technical replicates? How many true replicates were performed for this experiment (and for other related experiments)?

    Values were obtained from duplicate wells and are expressed as the mean ± SD. This sentence was added to the legends of Figures.